# The Missing Pieces to the Cold-Stored Platelet Puzzle

**DOI:** 10.3390/ijms23031100

**Published:** 2022-01-20

**Authors:** Hanqi Zhao, Dana V. Devine

**Affiliations:** 1Department of Pathology and Laboratory Medicine, University of British Columbia, Vancouver, BC V6T 1Z7, Canada; waynewainzhao@gmail.com; 2Centre for Blood Research, University of British Columbia, Vancouver, BC V6T 1Z3, Canada; 3Centre for Innovation, Canadian Blood Services, Ottawa, ON K1G 4J5, Canada

**Keywords:** cold-stored platelets, product storage, storage solution, platelet transfusion

## Abstract

Cold-stored platelets are making a comeback. They were abandoned in the late 1960s in favor of room-temperature stored platelets due to the need for longer post-transfusion platelet recoverability and survivability in patients with chronic thrombocytopenia. However, the current needs for platelet transfusions are rapidly changing. Today, more platelets are given to patients who are actively bleeding, such as ones receiving cardiac surgeries. It has been established that cold-stored platelets are more hemostatically effective, have reduced bacterial growth, and have longer potential shelf lives. These compelling characteristics led to the recent interest in bringing back cold-stored platelets to the blood systems. However, before reinstating cold-stored platelets in the clinics again, a thorough investigation of in vitro storage characteristics and in vivo transfusion effects is required. This review aims to provide an update on the recent research efforts into the storage characteristics and functions of cold-stored platelets using modern investigative tools. We will also discuss efforts made to improve cold-stored platelets to be a better and safer product. Finally, we will finish off with discussing the relevance of in vitro data to in vivo transfusion results and provide insights and directions for future investigations of cold-stored platelets.

## 1. Introduction

Cold-stored platelets (CPs) are not a new product. Prior to the late 1970s, all platelet units used for transfusions were stored at 4 °C. However, it was recognized in 1969 that CPs were quickly cleared from circulations, whereas room-temperature stored platelets (RPs) had significantly longer circulation time [1].

The rapid clearance of CPs is attributed to one of the significant physiological changes when platelets are exposed to the cold temperature. It was discovered that cold storage induces the clustering of platelet surface glycoprotein GPIbα, which has been shown to mediate phagocytosis by macrophages in short-term (<48 h) CPs [2,3]. Long-term cold-stored platelets (>48 h) are cleared by hepatocytes in the liver through interaction of the Ashwell–Morell receptor with the exposed β-GlcNAc moieties on CPs [4]. Efforts to prevent the rapid clearance of CPs by galactosylation of the β-GlcNAc moieties have not been successful in humans [5]. Since the majority of platelet transfusions were given to patients with hematological disorders or thrombocytopenic patients in the late 1970s, there was a complete shift in platelet storage methodology from CPs to RPs.

In addition to changes in surface receptors, it is observed that platelets stored in the cold experience significant morphological changes compared to fresh platelets. In the circulation, non-activated platelets have a thin discoid shape. Upon exposure to 0 °C for 10 min ex vivo, the majority of the platelets lose their discord shape and become spherical [6]. Under the electron microscope, platelets exposed to the cold are spherical with many “bumps” and often possess thin pseudopods extending outward. Furthermore, platelets exposed to the cold also loses their circumferential microtubule rings [7].

Cold temperature also induces activation of the platelets in the forms of platelet granule content release and the exposure of phosphatidylserine [8]. One possible mechanism of cold-induced platelet activation is the significant accumulation of intracellular calcium during cold storage [9]. 

Furthermore, cold temperature has significant effects on the metabolism of stored platelets. CPs have reduced glucose consumption and lactate production [10]. Storage in the cold has also been associated with preservation of the mitochondrial functions of the platelets [11].

Despite the rapid in vivo clearance, there has been a renewed interest in bringing back CPs as a product. This recent resurgence can be attributed to three main reasons. First, cold storage significantly reduces the growth of microorganisms, such as bacteria, in the platelet units [12]. This allows the platelet units to have a longer shelf life, thus reducing the wastage due to outdating. Second, CPs are shown to be hemostatically superior to RPs [13]. Lastly, while RPs require constant mechanical agitation during storage, CPs do not seem to require agitation [14]. This eliminates the cost of mechanical shakers and improves logistics handling during platelet shipping. Furthermore, recent investigations into the utilization of platelets for transfusion have shown that there is a significant increase in the need for therapeutic platelet transfusions compared to prophylactic transfusions [15,16]. With superior hemostatic functions and longer potential shelf life, CPs could be a promising product for actively bleeding patients.

Previous reviews have thoroughly discussed the in vitro characteristics, physiology, and hemostatic effectiveness of CPs [17,18,19]. A brief overview of the known characteristics of CPs and the gap in knowledge are summarized in Figure 1. This review aims to first provide an update on new findings of CPs generated by innovative in vitro models and advanced analytical techniques. Then, we will summarize the recent efforts to improve CPs quality through supplementation with antioxidants as well as making CPs safer with application of pathogen inactivation technology in conjunction with platelet additive solutions. Then, we will discuss the recent changes in CPs storage methodology. Finally, the review will finish with a discussion of the relevance of in vitro and in vivo data of CPs.

## 2. CPs In Vitro Characterization: An Update and a Debate

The common theme of the in vitro characterization studies in the past was that CPs had significantly higher hemostatic potential compared to RPs. During the late 1960s when the platelet storage practice was switched from CPs to RPs, it was known that CPs were better for bleeding patients, as they showed superior correction in bleeding time [13]. The latest in vitro models have revealed new information about the storage characteristics of CPs. Scanning electron microscopy was used to examine the characteristics of clots formed by CPs versus RPs [20]. The microscopy images suggested that CPs had significantly more crosslinking with denser fibers compared to clots formed by RPs. In vitro analysis of CPs stored for 5 days also showed similar clot stiffness to that of fresh platelets. The superior fiber crosslinking in CPs could be mediated by factor XIII through binding on the activated glycoprotein IIb/IIIa on the CPs surfaces [20]. In addition, clot histology showed that CPs had higher porosity and more well-defined structure in their clots similar to fresh platelets [21]. On the other hand, RPs formed clots with altered structure and decreased porosity. 

Mitochondria in platelets are essential to platelet energy metabolism and production [22]. Mitochondria also play an important role in mediating platelets’ response to stress and activation [22]. Reactive oxygen species (ROS), which are generated during mitochondrial respiration, were found to be lower in CPs compared to RPs [11]. Furthermore, CPs maintained mitochondrial respiration capabilities whereas RPs had decreased mitochondrial respiration after 7 days of storage [11]. Finally, cold storage has also been discovered to induce significant changes on the ADP receptors of platelets [23]. In particular, P2Y1 and P2X1 receptors decreased in CPs compared to RPs. The changes in the ADP receptors of CPs could provide an explanation as to why CPs are less responsive to inhibitory signals, such as nitric oxide and prostaglandin E1, compared to RPs [23].

The recent in vitro investigation of CPs also revealed contradictory data with regard to the superiority of the CPs hemostatic functions. For instance, it was shown that the concentration of glycoprotein VI (GPVI), the receptor on the platelet surface for collagen activation, was lower in CPs versus RPs [24]. Subsequently, using a custom microfluidics model designed to measure platelet contractile force under shear stress, CPs were found to have weaker aggregation response to collagen than RPs. In vivo transfusion experiments using human volunteers under dual antiplatelet therapy also showed that post-transfusion CPs responded less to collagen compared to RPs, further suggesting the loss of GPVI during storage in vitro [24]. However, it is important to note that these experiments were performed with washed platelets, which may affect GPVI level. There was also a limited number of biological replicates conducted in the human study (*N* = 8).

In addition, there are concerns about whether in vitro assays are reliable in studying CPs. One such example is the use of viscoelastic testing. Viscoelastic tests, such as rotational thromboelastometry (ROTEM) or thromboelastography (TEG), have often been used to show the superior hemostatic effectiveness of CPs over RPs. However, due to the significant changes in the surface receptors, such as glycoprotein IIb/IIIa in CPs, the results of viscoelastic testing on CPs may be misleading. When analyzing CPs on TEG under MA_fibrin_ mode, which was only supposed to reflect the contribution of fibrinogen to clots, the activated glycoprotein IIb/IIIa on CPs contributed to an increase in the maximum amplitude [25]. Blocking the glycoprotein IIb/IIIa using inhibitors eliminated this effect [25]. These data showed that the interpretation of CPs in vitro results should be cautious, and other assays should be used in tandem.

## 3. “Omics” and Cold-Stored Platelets

Recently, “omics” technology, such as proteomics and metabolomics, have become more accessible. This is facilitated by the decreased cost of performing these assays and the increased recognition that a large repertoire of information about platelet storage biology can be generated [26]. An untargeted metabolomic approach was used to examine the effect of cold temperature on metabolites during platelet storage. Compared to RPs, CPs had significantly decreased oxidative stress compared to RPs. CPs had reduced glutathione consumption during storage as well [14,27]. When examining the trends in metabolite changes using principal component analysis, the metabolomic profile of CPs stored for 21 days clustered closely to day 5 stored RPs [27]. Intuitively, the cold temperature should slow down the metabolism of platelets, which would lead to less activated metabolomic pathways; however, the metabolomic data suggested otherwise. Some glycolytic pathways in CPs, such as the pentose phosphate pathway and the purine metabolism pathway, had an increase in activity after prolonged storage in the cold [27]. Results from untargeted metabolomics were also used to complement the in vitro characterization of stored platelets. Specifically, metabolomic pathways that respond to oxidative stress, such as the pentose phosphate pathway, were shown to positively correlate with platelet function measured by ROTEM and light transmission aggregometry [14]. In addition, mechanical shaking did not result in a significant difference in the in vitro storage characteristics or metabolomic profiles of CPs [14]. This suggests that agitation may not be required in CPs.

Changes in the platelet proteomes during cold storage were also examined. Different storage temperatures of platelets led to differentially expressed proteins important in platelet degranulation, blood coagulation vesicle transport, protein activation cascade, and response to stress [28]. Interestingly, many of the differentially expressed proteins were located in the membrane-bound vesicles of platelets. Some examples of these differentially expressed proteins were GP5, TBXAS1, and FERMT3. These proteins have been found to have a role in regulating platelet activation during storage [29,30]. Storage temperature also had a significant impact on the microRNA (miRNA) expression profile of platelets. In a preliminary study, it was found that there were over 100 differentially expressed miRNAs between fresh platelets, CPs, and RPs. Furthermore, qPCR confirmed that several miRNAs, such as mir-20a-5p, had significantly increased expression in CPs compared to RPs [31]. The significance of these miRNA correlation with platelet quality is still being investigated.

## 4. Reduction of CPs Storage Lesion, a Work in Progress

Results from the recent omics study showed that oxidative stress played a significant role in platelet storage lesion both in RPs and CPs. Hence, to improve CPs’ storage quality, both synthetic and naturally occurring antioxidants have been used as supplements. N-acetylcysteine (NAC) has been used as a supplement to minimize the mitochondrial uncoupling and proton leaks caused by cold-storage [32]. NAC supplement also showed the slight effect of reducing ROS in CPs. Both RPs and CPs with NAC supplement had better correction in a mice tail bleeding experiment than CPs without NAC [32]. However, these experiments had relatively few biological replicates. In addition, the use of platelet additive solution (PAS) could have influenced the results as well. In another study, whole blood-derived platelet rich plasma (PRP) supplemented with NAC and stored in the cold seemed to have reduced platelet activation measured by P-selectin expression compared to RPs [33]. NAC supplement did not result in differences in the phosphatidylserine expression of CPs or RPs. However, the platelets used in this experiment were stored in gas-permeable tubes instead of standardized storage containers, making it difficult to draw solid conclusions [33]. In a follow-up study, the effect of NAC on CPs post-transfusion recovery has been tested using the Prkdc^scid^ mouse model [34]. There was a slight improvement of platelet recovery 2 h post transfusion in the CPs with the NAC group. However, this improved recovery effect was diminished after 24 h post transfusion [34]. It is important to note that in this mouse study, CPs without NAC supplement were not tested due to aggregate formation in the storage bag. The absence of this control makes data interpretation more challenging, as the result could be due to either the storage temperatures or the addition of NAC. Another antioxidant, resveratrol, and a mitochondrial targeting protein, cytochrome c, have been used to reduce oxidative stress in CPs as well [35]. The preliminary results from these supplements showed decreased ROS in resveratrol-treated CPs [35]. However, further investigation of supplemented CPs function and transfusion efficacy are needed.

In addition to using antioxidants, synthetic molecules have also been used to reduce the lesion in CPs. Specialized pro-resolving mediators (SPMs) are biosynthesized molecules derived from essential fatty acids. SPMs have been found to reduce acute inflammation during the host response to invading pathogens [36]. Specific SPMs, such as resolving E1, have been found to regulate platelet activations [37]. In CPs, the addition of SPMs significantly reduced the glycoprotein 1b expression compared to CPs without SPMs [38]. However, in this study, there were no RPs with SPMs as a control. It is not clear what other effects SPMs may have in CPs during storage. Similarly, sodium octanoate (SO) was used as a supplement in CPs to improve mitochondrial activity and reduce CPs apoptosis [39]. Although CPs with SO seem to trend toward better mitochondrial activity compared to CPs without SO, the results were not statistically significant. Furthermore, although a lower phosphatidylserine expression was observed with CPs with SO on day 5 of storage, further analysis of the caspase-3 level between the treatment groups was not significant. Platelet endocytosis with HepG2 cell models also did not show a difference in SO-treated versus non-treated CPs [39]. Finally, integrin inhibitor RGDS and a calcium ion chelator EGTA were added to improve cold-stored platelet post-transfusion clearance [40]. It is important to note that this experiment was performed with mouse platelets, and the cold storage period of murine platelets was for a maximum of 24 h. While the calcium chelator EGTA showed some improvement in cold-stored murine platelet clearance, the effect disappeared when EGTA was used in human platelets. In addition, EGTA supplementation in human PRP prevented the reduction in platelet count commonly observed in CPs during storage as well as aggregate formation. However, CPs stored in PAS could also achieve similar effects, so the exact function of EGTA in CPs is not as straightforward as it seems.

## 5. Safer CPs: Pathogen Inactivation and Platelet Additive Solutions

Despite stringent donor eligibility criteria, blood products are still exposed to the risk of emerging pathogen contaminations. West Nile outbreaks in the past and the recent outbreak of Zika virus in the US are good examples of this [41,42]. Pathogen inactivation (PI) technology has been attracting interest in further securing the safety of blood components [43]. This technology targets and damages the nucleic acids of pathogens using ultraviolet light, thus preventing pathogen replication in blood products. PI RPs have been licensed in many countries including the US, Spain, and Switzerland, according to a report by AABB [44]. CPs, due to the lower storage temperature, have been shown to slow the growth of bacteria [12]. However, psychrophilic bacteria, such as *Yersinia enterocolitica,* can survive at 4 °C and proliferate to a clinically significant level [45]. *Yersinia enterocolitica* has been associated with transfusion reaction in red blood cell transfusions [46]. Thus, there are incentives in investigating the effect of PI on CPs. Recent data show that CPs in PAS treated with Intercept PI (Cerus Corporation, Concord, CA, USA) had less activation level and aggregation in response to agonists than non-PI treated CPs [47]. However, when tested in microfluidic devices, PI-treated CPs had a better fibrin formation rate [47]. One caveat for this study is that the platelets were only stored for up to 5 days in the cold. When Intercept PI-treated CPs were stored for an extended period up to 21 days, there was no major difference in the metabolism of the platelets in terms of count, pH, glucose, or lactate compared to non-PI-treated CPs [48]. Interestingly, PI treatment resulted in significant changes to CPs function. There was a significant reduction in clot retraction capabilities in PI-treated CPs. There was also elevated phalloidin staining in PI-treated CPs. Furthermore, there was also an elevated lysis index at 30 min measured by ROTEM in PI-treated CPs [48]. These data suggest that PI may adversely affect CPs during storage. A different PI system, Theraflex UV-platelet System (UVC), was also used on CPs [49]. Similar to the effects of the Intercept system, this technology also resulted in damage to the CPs, as shown by decreased HSR response. In addition, PI-treated CPs also had higher activation shown by annexin V staining and higher release of microparticles compared to untreated CPs [49]. Together, these results show that PI does have significant effects on the cold storage of platelets. More data on the post-transfusion effectiveness of PI-treated CPs are needed to fully understand the feasibility of these platelets in the clinic.

Platelet additive solution (PAS) is a chemical solution that can replace plasma in platelet storage. Different types of PAS used in studying CPs have been summarized in Table 1. PAS provides several benefits. First, PAS can be used to replace some of the plasma in platelet storage. Then, the replaced plasma can be used for fractionation into intravenous immunoglobulin (IVIg) to help with the ongoing shortages [50,51]. In addition, reducing the amount of plasma in a platelet unit lowers the risk of allergic transfusion reactions [52]. This adds an additional layer of safety for the transfusion recipients. In CPs, there have been suggestions that PAS could be beneficial [53]. It was first established that similarly to PAS in RPs, a residual of 35% plasma is optimal for CPs in stored PAS [54]. Based on this result, the effects of different PAS on CPs storage quality have been investigated. For example, two FDA-approved collections and PAS storage systems of platelets (Trima apheresis collection system in Isoplate PAS versus Amicus collection system in Intersol PAS) stored in the cold were compared [55]. The results showed that there were only minor differences in the platelet count over 15 days of storage between the two systems. CPs produced from the two systems also had similar aggregation response profiles [55]. It is important to note that the minor differences observed could be contributed by either the different apheresis collection instruments or by the different PAS. To further differentiate the storage quality of CPs in PAS versus in plasma, ISO PAS platelets were stored for 19 days and compared to apheresis platelets in 100% plasma [56]. CPs stored in PAS had significantly reduced glucose consumptions versus CPs in plasma. This is likely due to the substitution of acetate as an energy source instead of glucose, while also reducing lactate production. Day 19 CPs PAS also had acceptable pH, which would meet FDA pH criteria [56]. One criticism for this work is that the CPs in 100% plasma were only stored for 3 days, making parallel comparison to day 19 CPs in PAS difficult. CPs stored in PAS-E in also maintained pH above 6.4 for 21 days [57]. The aggregation response to agonists is the same between CPs in PAS versus CPs in plasma. However, it is important to note that the addition of PAS to CPs reduced fibrinogen concentration and other coagulation factors, which could be detrimental to the recipient upon transfusion. In addition, PAS could provide additional benefits specific to CPs. The lowered fibrinogen concentration in PAS leads to a significant reduction of aggregate formation while preserving the in vitro function in CPs [10]. Additionally, PAS may help in maintaining mitochondrial integrity by reducing the depolarization of the inner membrane potential in CPs [58]. Finally, the type of PAS used in CPs could also affect the post-transfusion recovery. The recoveries of CPs stored in either 100% plasma, Intersol, or Isoplate PAS were compared in an in vivo transfusion of autologous radiolabeled platelets into healthy donors. Interestingly, CPs stored in Intersol had better recoveries than CPs in Isoplate. However, CPs stored in 100% plasma still had better than CPs stored in either PAS [59].

## 6. Changes in Cold Storage Methodology Paradigm: Delayed Cold Storage and Temperature Cycling

One benefit of CPs is that we can alleviate the platelet wastage rate due to the CPs’ longer shelf life. However, having two platelet inventories to manage adds complexity to the blood operators and hospitals. There are already many challenges associated with managing the current platelet inventories [62,63]. One way to mitigate these problems would be to store platelets at room temperature until close to the end of their shelf life and then transferring them to the cold with the hope to extend their viability and reduce platelet wastage. This idea is known as the delayed cold storage of platelets. The possibility of delayed storage and the quality of delayed cold stored platelet product have been assessed. In a recent study, platelets in PAS were split into two identical units. One unit was immediately stored at 4 °C after production, while the other unit was stored at room temperature for 4 days prior to being transferred to 4 °C. Both units were tested for in vitro parameters for up to 21 days [60]. It was shown that delayed CPs had similar count and mean platelet volume compared to the CPs. However, PAS was likely the main contributor here. Intuitively, due to first being stored at room-temperature, the delayed CPs had worse blood gas parameters consisting of higher glucose consumption, lactate production, and lower pH compared to platelets that were stored in the cold immediately after production. Interestingly, delayed CPs had better hypotonic shock response recovery compared to CPs, suggesting a potential benefit of delayed cold storage [60]. Similarly, compared to CPs, apheresis platelets in PAS that were transferred to the cold after 7 days at room temperature showed worse in vitro parameters [61]. Furthermore, platelet functional analysis by TEG and aggregometry showed that delayed CPs had worse clot dynamics and aggregation response to agonists, respectively [61]. Although delayed cold storage looks to be the best of both worlds in terms of reducing platelet wastage while not adding the extra complexity of a dual inventory, the quality and usability of delayed CPs need to be carefully examined.

In addition to delayed cold storage, another change in the CPs storage method that may improve quality is temperature cycling (TC). This idea is based on the findings that the continuous cold storage of platelets is deleterious to platelet storage quality [64]. By alternating between cold and room temperatures during storage, the platelets would have the opportunity to recover at warmer temperatures and partially reverse the changes caused by cold storage [65]. When platelets were cycled between 4 °C storage for 12 h followed by 37 °C storage for 30 min, the platelets showed superior recovery in an SCID mouse model compared to non-TC RPs or CPs [66]. To reduce the logistic efforts of manual temperature cycling, an automated temperature cycling machine that cycles between 11 h of 5 °C storage followed by 1 h of recovery at 22 °C was used to perform further studies. In the in vitro analyses, TC platelets showed better storage characteristics compared to CPs but worse than RPs. When the TC platelets were tested using the SCID mouse model of transfusion, they showed comparable recovery to RPs. CPs showed the worst post-transfusion recovery [67]. Then, the in vivo recovery and survival of TC platelets were compared to RPs and CPs in healthy human volunteers. In humans, the results were not as optimal as they were in the mouse model. TC platelets transfused to humans showed recovery better than CPs but worse than that of RPs [68]. A P38 mitogen-activated protein kinase inhibitor (VX-702) has been used to try to improve the TC platelets. This inhibitor seems to improve the in vitro storage characteristics of TC platelets but not CPs. When used in an SCID mouse transfusion model, the addition of the inhibitor to TC platelets initially had better recovery than TC without the inhibitor. However, the effect was temporary and quickly diminished [69]. While the idea of temperature cycling is attractive, it requires additional equipment implementation, which could be costly. The further investigation of post-platelet transfusion effectiveness, possible increased risk of bacterial growth, and hemostatic efficacy of TC platelets are also needed to fully understand the utility of TC platelets.

## 7. CPs in the Clinics: Regulatory Approval, Application, and Where In Vitro Analyses Fit In

The inconsistent correlation of in vitro data to in vivo results has always been one of the most difficult obstacles in studying platelet storage [70]. The relevance of many in vitro analyses of stored platelets to in vivo effects has often been questioned [71]. One such example recently came from the investigation of CPs in restoring vascular endothelial functions. When added to an in vitro leaky endothelial cell culture model, CPs had superior reductions of endothelial cell permeability compared to RPs. However, when tested in a mouse model, the superior effect of CPs diminished compared to RPs upon transfusion into the mice. CPs still provided significant leak reduction compared to non-transfused mice, but the effect was not comparable to RPs-transfused mice [72]. When investigating the procoagulant subpopulation of platelets, defined as CD62P^+^ and GSAO^+^ by Hua et al. [73], it was initially discovered that while cold storage increases the proportion of procoagulant platelets, their response in vitro to agonists stimulation seems to be blunted compared to fresh in vitro. However, when these platelets were transfused, their response to agonist stimulation was restored [74].

This does not mean that investigations of the in vitro storage characteristics of CPs are meaningless. Perhaps focusing on in vivo results using radiolabeled platelets transfusion into healthy donors is not the best way to assess the accuracy of in vitro results [75]. Since CPs have the greatest potential to be a superior product for actively bleeding patients, in vivo results from therapeutic transfusion using CPs should be prioritized.

However, it has been challenging to implement CPs in the clinics due to the process of regulatory approval. In the Mayo Clinic in Rochester, Minnesota, CPs have been utilized for bleeding patients since 2015 following the FDA outline of a maximum storage duration of 3 days. Due to this regulation, there was a large discard rate, and only 19.1% of the collected CPs were transfused [76]. This center has been trying to obtain regulatory approval from both the AABB and FDA since 2013 to store apheresis platelets in the cold for 5 days without agitation. However, this goal was not achieved, and the FDA only approved 3-day cold storage without the need for bacterial testing in the end [76]. Recently, during the early periods of the COVID-19 pandemic, there were serious concerns about blood product shortages [77,78]. Given the short storage life of RPs, this has pushed FDA to grant several emergency approvals for extending product shelf life using CPs. In February 2020, the FDA has granted the Texas Blood & Tissue Center license to make and distribute platelets stored in the cold for 14 days. In addition, in Mayo Clinic Rochester, pathogen-inactivated RPs after 5 days of storage were allowed to be transferred to cold storage for an additional 9 days [79]. A total of 61 units of these delayed cold-stored platelets were transfused to 40 patients. The majority of these patients were receiving cardiac surgeries. The results from these transfusions showed that the transfusion recipients had adequate surgical hemostasis comparable to patients receiving RPs. There were also no documented transfusion reactions related to the cold-stored platelet transfusion [79]. More evidence from a pilot clinical trial of CPs further demonstrates the feasibility of cold-stored platelets in treating bleeding during cardiothoracic surgeries. In adults receiving complex cardiothoracic surgery, the median chest tube drain output of patients who received day 14 CPs was similar to the median chest tube drain output of patients receiving day 7 RPs. Secondary outcomes such as platelet function, total blood usage, immediate and long-term adverse events, length of stay in intensive care, and mortality were comparable between patients receiving the two different methods of stored platelets [80].

## 8. Future Directions and Conclusions

As CPs gains increased research interest, it is important to keep in mind that as a collaborative research community, we should work together to eliminate as many confounding factors as possible in the preparation and storage of CPs. This will ensure the reliable and replicable results for the use of CPs. In addition, clinical trials involving acute bleeding patients receiving CPs compared to patients receiving RPs are crucial in moving forward with the approval of CPs in the blood systems. These efforts are already underway, as there have been publications discussing clinical trial designs in investigating CPs [81]. Finally, the currently ongoing phase 3 randomized clinical trial headed by Dr. Philip Spinella called the Chilled Platelet Study “CHIPS” should generate exciting results for the future studies of cold-stored platelets.

## Figures and Tables

**Figure 1 ijms-23-01100-f001:**
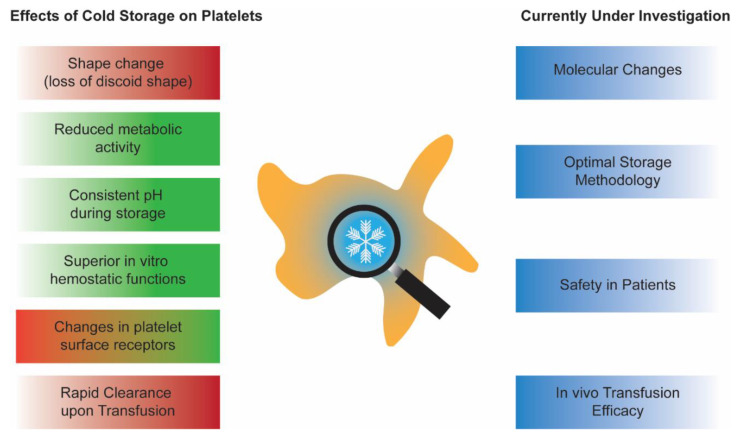
Summary of known effects of cold storage on platelets and aspects of cold-stored platelets currently under investigation. Deleterious effects of cold storage are shaded in red, whereas beneficial effects are shaded in green. The box shaded with red and green colors represents conflicting effects of cold storage on platelet surface receptors.

**Table 1 ijms-23-01100-t001:** Different platelet additive solutions (PAS) used in recent investigations of cold-stored platelets (CPs).

PAS	Author	Ref	Main Usage
T-PAS	Nair et al.	[21]	Investigation of the clot structure of CPs.
T-PAS	Reddoch-Cardenas et al.	[56]	Compared CPs stored in PAS versus RPs and CPs in plasma.
T-PAS and Intersol	Getz et al.	[10]	Reduce aggregate formation in CPs using PAS.
SSP+	Six et al.	[47]	Pathogen inactivation (Intercept).
SSP+	Johnson et al.	[49]	Pathogen inactivation (Theraflex UV-platelet System).
SSP+	Marini et al.	[54]	Plasma replacement and test for the percentage of residual plasma required.
SSP+	Wood et al.	[60]	Delayed cold storage.
PAS-E	Hegde et al.	[32]	NAC supplementation in CPs
PAS-E	Johnson et al.	[57]	Prolonged platelet cold storage.
Intersol	Agey et al.	[48]	Pathogen inactivation (Intercept).
Isoplate	Reddoch-Cardenas et al.	[58]	Preserving mitochondrial function in CPs.
Isoplate and Intersol	Reddoch-Cardenas et al.	[55]	Compare the CPs storage characteristics in two FDA approved PAS.
Isoplate and Intersol	Stolla et al.	[59]	PAS in CPs in vivo recoveries using autologous radiolabeled platelet transfusion.
PAS-IIIM	Braathen et al.	[61]	Delayed cold storage.

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
