# Peer review of "The Missing Pieces to the Cold-Stored Platelet Puzzle"

_ijms, 2022, doi:10.3390/ijms23031100_

Round 1

Reviewer 1 Report

This is an interesting review paper about cold-stored platelets and their potential use in clinical aspects. Unfortunately, during the first reading, the manuscript gives the impression of being a bit chaotic and not well organized, with poorly characterization of cold stored platelets in vitro. The description of changes in ultrastructure were almost completely omitted, if the Figure 1 and the cited literature are not taken into account. It is true that the authors mention in the introduction "A brief overview of the known characteristics of CPs .... are summarized in Table 1", but they rather meant Figure 1. Only after reading a few other articles on "cold storage of platelets "and rereading this manuscript, it is easier to understand the message of reviewed paper. The authors did not avoid repetition by providing the same information in different workplaces. I also find it unnecessary to insert in section 6 (Change in Cold Storage Pardigim ...) that "In later section of this review, we will discuss an example of delayed CPs transfussion into cardiac surgery patients". In my opinion, it is not needed here, and if the authors want to mention the use of CPs, they should describe it in one place. It is worth checking the text for typos. For example, on page 11, the authors meant 2020 rather than 2022.
Overall, this article is interesting, only the presentation of individual issues is somewhat uncoherent. Perhaps authors should consider a slight correction / reorganization to increase its readability. I admit that much more useful information, in much less time,  I got by reading "Hastings H, Cancelas JA. Cold Stored Platelets. PathologyOutlines.com website. https: www.pathologyoutlines.com/topics/transfusionmedcoldstoredplatelets.html" than with the reviewed manuscript.
In my opinion, the article is worth publishing after minor corrections.

Author Response

Thank you for your helpful comments.  We have done some reorganizing of the paper to improve its readability.   We have also corrected the typos and grammatical errors.

Reviewer 2 Report

This review by Zhao et al, on The Missing Pieces to the Cold-stored Platelet Puzzle is well developed and is of great current clinical interest on hemostasia.

Author Response

Thank you for your support of our manuscript.